# Highly Sensitive Detection of Microstructure Variation Using a Thickness Resonant Transducer and Pulse-Echo Third Harmonic Generation

**DOI:** 10.3390/ma16134739

**Published:** 2023-06-30

**Authors:** Hyunjo Jeong, Hyojeong Shin, Shuzeng Zhang, Xiongbing Li

**Affiliations:** 1Department of Mechanical Engineering, Wonkwang University, Iksan 54538, Republic of Korea; 2Graduate School of Flexible and Printable Electronics, Jeonbuk National University, Jeonju 54896, Republic of Korea; hjshin95@jbnu.ac.kr; 3School of Traffic and Transportation Engineering, Central South University, Changsha 410083, China; sz_zhang@csu.edu.cn (S.Z.); lixb213@csu.edu.cn (X.L.)

**Keywords:** harmonic generation, thickness resonance, cubic nonlinear parameter, source nonlinearity, precipitation heat treatment

## Abstract

In nonlinear ultrasound testing, the relative nonlinear parameter is conveniently measured as a sensitive means of detecting and imaging overall variation of microstructures and damages. Compared to the quadratic nonlinear parameter ($\beta^{\prime }$), the cubic nonlinear parameter ($\gamma^{\prime }$), calculated as the third harmonic amplitude divided by the cube of the fundamental amplitude, has generally a higher value, providing better sensitivity in nonlinear parameter mapping. Since the third harmonic amplitude is about two orders of magnitude lower than the fundamental amplitude, efficient excitation and highly sensitive reception of third harmonic is very important. In this paper, we explore an odd harmonic thickness resonant transducer that meets the requirements for pulse-echo third harmonic generation (THG) measurements. We also address the problem of source nonlinearity that may be present in the measured amplitude of the third harmonic and propose a method to properly correct it. First, we measure $\gamma^{\prime }$ for a series of aluminum specimens using the through-transmission method to observe the behavior of $\gamma^{\prime }$ as a function of specimen thickness and input voltage, and examine the effects of various corrections such as attenuation, diffraction and source nonlinearity. Next, we apply the odd harmonic resonant transducer to pulse-echo THG measurements of precipitation heat-treated specimens. It is shown that such transducer is very effective in generation and detection of fundamental and third harmonics under finite amplitude toneburst excitation. The highly sensitive detectability of $\gamma^{\prime }$ are presented as a function of aging time, and the sensitivity of $\gamma^{\prime }$ is compared with that of $\beta^{\prime }$ and ${\beta^{\prime }}^{2}$.

## 1. Introduction

Nonlinear ultrasound testing (NLUT) is a type of nondestructive evaluation technique that diagnoses material conditions and defect characteristics by examining harmonics generated when a longitudinal wave of frequency *f* of finite amplitude propagates in a material. Harmonics of frequency 2*f* or 3*f* are frequently used. NLUT has been found to be more sensitive to microdamage and contact nonlinearity than conventional linear ultrasound, and its application is gradually increasing. Quadratic and cubic nonlinear parameters are frequently measured as sensitive means to detect and image changes in microstructure and damage. Compared to the quadratic nonlinear parameter ($\beta^{\prime }$), the cubic nonlinear parameter ($\gamma^{\prime }$) which is calculated as the third harmonic amplitude divided by the cube of the fundamental wave amplitude has a higher value, thus providing better sensitivity. Measurements of $\gamma^{\prime }$ for fatigue cracks [1,2], plastic deformation [3,4,5] and microstructures [6,7,8] have been reported in the literature. 

Most third harmonic generation (THG) measurements of longitudinal waves so far have been made in the through-transmission testing mode. If cubic nonlinear parameters can be measured through the pulse-echo THG, it will be of great help to the field application of NLUT. However, there are several factors to consider when developing a pulse-echo THG measurement technique. Considering a single element transducer acting as a transmitter and receiver in pulse-echo test, such transducer cannot be used for second harmonic generation (SHG) and reliable measurement of β of solids due to the very weak amplitude of the second harmonic arriving at the receiver position. The second harmonic behavior at the stress-free reflecting boundary can be described as follows based on the quasilinear theory of nonlinear wave equations [9,10], the solution to the second harmonic in the reflected beam consists of two separate contributions. One is the reflected second harmonic which is generated in the forward propagation path, while the other one is the newly generated by the reflected fundamental in the backward propagation path. Because of the phase difference of π between the two components, the second harmonic completely cancels out after reflection from the stress-free boundary. For this reason, most second harmonic generation measurements have been limited to through-transmission methods. 

Unlike the behavior of second harmonics [9,10], the behavior of the third harmonic at the stress-free boundary can be treated like a continuously transmitted wave propagating twice the sample thickness because there is no phase difference between the two third harmonics after reflection [11,12]. The resulting third harmonic component accumulates in proportion to the propagation distance. Therefore, the pulse-echo THG measurement can be realized.

The intensity of higher harmonic amplitudes generated from damaged materials is very weak compared to that of the fundamental amplitude. The second harmonic is generated at an amplitude of about 20 dB lower than that of the fundamental wave, and the third harmonic is generated at an amplitude that is about 20 dB lower than that of the second harmonic. Therefore, efficient excitation and highly sensitive detection of the third harmonic is quite important, and transducers play an important role in NLUT systems for THG measurements. For transmit transducers, higher harmonics can be effectively generated by exciting a single crystal piezoelectric element such as lithium niobate (LiN) with a narrowband toneburst at the fundamental frequency [13,14,15]. The receive transducer should be able to receive the third harmonic generated by the target material with reasonable sensitivity. Therefore, receiver bandwidth is important. Limited receiver bandwidth results in reduced detection of third harmonic amplitude, leading to underestimation of the cubic nonlinear parameter. In THG measurements, a broad bandwidth receiver that covers both fundamental and third harmonic frequencies was used [16]. It is desirable that the frequencies of the fundamental and third harmonics be located within the 50% bandwidth. Ultrasonic receivers with double-peak-type frequency characteristics were designed and fabricated for third harmonic reception [17]. The peak at the low frequency side is used to receive the fundamental wave, and another peak at the high frequency side is used to receive the third harmonic. It is generally not easy for a single element transducer to meet the requirements of transmitter and receiver simultaneously. Separately arranged transmitter/receiver ultrasonic transducers were also used for third harmonic signal detection with high selectivity and high reception sensitivity in tissue harmonic imaging [18,19]. Similar dual element transducers were also used for pulse-echo second harmonic generation measurements [15,20]. 

Because third harmonic generation is about one order of magnitude lower than second harmonic generation, efficient excitation and highly sensitive detection are very important. In addition to broad bandwidth transducers and double-peak-frequency transducers mentioned above, a transmit/receive transducer that is sensitive at both fundamental and third harmonic frequencies was proposed [21] for medical ultrasound applications. These frequencies respectively correspond to the fundamental and third harmonic thickness resonances of the piezoelectric transducer. Although the thickness resonance of piezoelectric elements is well known, the nature of higher harmonic thickness resonances associated with NLUT has not been sufficiently investigated. Therefore, the first objective of this paper is to further explore this type of transducer that meets the requirements for the pulse echo NLUT, and apply it to highly sensitive detection of microstructural changes due to precipitation heat treatment. 

When a finite amplitude toneburst pulse of frequency *f* excites a piezoelectric element, it also resonates at the 3*f*, 5*f*, 7*f*, … overtones. Consequently, a large 3*f* component can be generated and propagates as another fundamental wave of 3*f*. This is called “source nonlinearity” and will add up with the third harmonic caused by damage. The amplitude of source nonlinearity can be large enough to mask the nonlinear third harmonic, so it is important to carefully check for its presence and properly remove it for accurate evaluation of $\gamma^{\prime }$. The second objective of current study is the development of a source nonlinearity correction method when an odd harmonic thickness resonant transducer is used for the pulse-echo third harmonic NLUT.

The paper is structured as follows. We first measure $\gamma^{\prime }$ for a series of aluminum specimens of different thicknesses using the conventional through-transmission method. We observe the behavior of $\gamma^{\prime }$ as a function of specimen thickness and input power, and examine the effects of diffraction correction, attenuation correction and source nonlinearity correction. In addition, we investigate the relationship between $\gamma^{\prime }$ and ${\beta^{\prime }}^{2}$, the square of the quadratic nonlinear parameter. Next, we apply the odd harmonic thickness resonant transducer to pulse-echo third harmonic generation (THG) measurements of precipitation heat-treated specimens. We will show that such transducer is very effective in generation and reception of fundamental and third harmonics under finite amplitude tone burst excitation. The procedure of source nonlinearity correction and its effect are explained in detail. We present highly sensitive detection results of $\gamma^{\prime }$ for heat-treated specimens subjected to different aging times. The sensitivity of $\gamma^{\prime }$ is compared with that of $\beta^{\prime }$ and ${\beta^{\prime }}^{2}$.

## 2. Absolute and Relative Nonlinear Parameters 

Consider a longitudinal wave propagation in an isotropic elastic medium with cubic nonlinearity. At any given time *t*, the displacement of a particle *x* from its initial position is described by the displacement *u*(*x*,*t*). The displacement equation of motion governing the longitudinal wave propagation can be deduced as [22]
(1)$$\frac{1}{c^{2}}\frac{\partial^{2}u}{\partial t^{2}}-\frac{\partial^{2}u}{\partial x^{2}}=-\left[\beta \left(\frac{\partial u}{\partial x}\right)+\gamma {\left(\frac{\partial u}{\partial x}\right)}^{2}\right]\frac{\partial^{2}u}{\partial x^{2}}$$
where β and γ are the quadratic and cubic nonlinear parameters, and *c* is the longitudinal wave speed. Assume a harmonic displacement boundary condition prescribed at x=0:(2)$$u\left(0,t\right)=U_{0}\sin\left(\omega t\right)$$
where $U_{0}$ is the initial source amplitude, ω is the angular frequency, and *k* is the wave number.

The perturbation technique can be applied to obtain the solutions to the boundary value problem given by Equations (1) and (2) [22]: (3)$$u=U_{0}\sin\left(kx-\omega t\right)+\frac{\beta U_{0}^{2}k^{2}x}{8}\cos2\left(kx-\omega t\right)+\frac{\beta^{2}U_{0}^{3}k^{4}x^{2}}{32}\sin3\left(kx-\omega t\right)$$

The amplitude of the fundamental is $U_{1}=U_{0}$ in Equation (3), and the amplitude of the second harmonic is $U_{2}=\frac{\beta U_{0}^{2}k^{2}x}{8}$. The amplitude of the third harmonc is obtained as $U_{3}\approx \frac{\beta^{2}U_{0}^{3}k^{4}x^{2}}{32}$ when $\beta^{2}$ dominates the third harmonic [23]. In this study, $\beta^{2}$ will be denoted by γ for convenience. 

When finite size transducers are used for generation and reception of ultrasound beam in a lossy medium, the amplitudes of the fundamental and higher harmonic waves are modified by attenuation and diffraction effects,
(4)$$U_{1}\left(x\right)=U_{0}\left[M_{1}\left(x\right)\right]\left[D_{1}\left(x\right)\right]$$
(5)$$U_{2}\left(x\right)=\frac{\beta U_{0}^{2}k^{2}x}{8}\left[M_{2}\left(x\right)\right]\left[D_{2}\left(x\right)\right]$$
(6)$$U_{3}\left(x\right)=\frac{\gamma U_{0}^{3}k^{4}x^{2}}{32}\left[M_{3}\left(x\right)\right]\left[D_{3}\left(x\right)\right]$$
where $M_{i}\;\mathrm{and}\;D_{i},\;i=1,2,3$ denote the attenuation and diffraction corrections at distance *x*.

We can now define the displacement-based or absolute nonlinear parameter. The second-order or quadratic nonlinear parameter β can be determined from Equations (4) and (5). The third-order or cubic nonlinear parameter γ can be determined from Equations (4) and (6).
(7)$$\beta \left(x\right)=\frac{8U_{2}\left(x\right)}{k^{2}xU_{1}^{2}\left(x\right)}\left[\frac{M_{1}^{2}\left(x\right)}{M_{2}\left(x\right)}\right]\left[\frac{D_{1}^{2}\left(x\right)}{D_{2}\left(x\right)}\right]$$
(8)$$\gamma \left(x\right)=\frac{32U_{3}\left(x\right)}{k^{4}x^{2}U_{1}^{3}\left(x\right)}\left[\frac{M_{1}^{3}\left(x\right)}{M_{3}\left(x\right)}\right]\left[\frac{D_{1}^{3}\left(x\right)}{D_{3}\left(x\right)}\right]$$

Equations (7) and (8) will be reduced to the definition of nonlinear parameters based on the pure plane wave solutions if the attenuation and diffraction corrections are neglected. Evaluation of absolute nonlinear parameters basically requires measurements of displacement amplitudes of fundamental and harmonically generated waves.

The relative nonlinear parameter can be determined more conveniently by using spectral peak values of the received electrical signal. If displacement amplitudes $U_{i},\;i=1,2,3$ in Equations (7) and (8) are replaced by spectral peak values $A_{i},\;i=1,2,3$, the relative nonlinear parameters $\beta^{\prime }$ and $\gamma^{\prime }$ are defned as
(9)$$\beta^{\prime }\left(x\right)=\frac{A_{2}\left(x\right)}{xA_{1}^{2}\left(x\right)}\left[\frac{M_{1}^{2}\left(x\right)}{M_{2}\left(x\right)}\right]\left[\frac{D_{1}^{2}\left(x\right)}{D_{2}\left(x\right)}\right]$$
(10)$$\gamma^{\prime }\left(x\right)=\frac{A_{3}\left(x\right)}{x^{2}A_{1}^{3}\left(x\right)}\left[\frac{M_{1}^{3}\left(x\right)}{M_{3}\left(x\right)}\right]\left[\frac{D_{1}^{3}\left(x\right)}{D_{3}\left(x\right)}\right]$$

Equations (9) and (10) are reduced to $\beta^{\prime }=A_{2}/A_{1}^{2}\;\mathrm{and}\;\gamma^{\prime }=A_{3}/A_{1}^{3}$ if attenuation and diffraction corrections are neglected and the propagation distance or sample thickness is constant.

## 3. Third Harmonic Generation Measurement and Effect of Various Corrections

Measuring relative cubic nonlinear parameters from THG in pulse-echo mode and correlating them with material conditions could be a new and promising NLUT technique with great advantages in practice. Before exploring a pulse-echo measurement technique, $\gamma^{\prime }$ was measured first using through-transmission method, and the influence of various corrections and the relationship with the quadratic nonlinear parameters are experimentally investigated to figure out properties of $\gamma^{\prime }$ and to validate the measured $\gamma^{\prime }$. Corrections considered here include attenuation correction, diffraction correction and source nonlinearity correction.

### 3.1. Materials and Experimental Setup

Commercial grade aluminum alloy 6061-T6 was used in this study. They were as fabricated and contained a basic material nonlinearity. Six specimens in the shape of a cuboid were prepared. The size of the cross section of each specimen is 4 cm × 4 cm. The lengths of the six specimens were 2, 4, 6, 8, 10, and 12 cm. The two surfaces in contact with the transducers were further processed to be smooth, flat and parallel to each other.

Harmonic generation measurements were conducted using a finite amplitude, through-transmission method. A single crystal lithium niobate (LiN) (5 MHz center frequency, 0.5 inch diameter) was used as a transmitter (T), and a broadband transducer of the same diameter was used as a receiver (R). For maximum capture of output signal, the two transducers were aligned coaxially in a pressurization fixture. 

The block diagram of the experimental setup is shown in Figure 1. A high power toneburst pulser (RPR-4000, RITEC, Warwick, RI, USA) was employed to produce a 5 MHz, 20 cycle toneburst input signal. A 150 Ohm high power feedthrough and a high power stepped attenuator were used. The receiver signal was captured on a digital storage oscilloscope (WaveSurfer 3024, Teledyne LeCroy, Chestnut Ridge, NY, USA). For each sample, input powers of nine levels (from 0 to 40 power levels in 5 level steps) were applied from the high power pulser. These input voltages fall within the range of about 30–300 $V_{\mathrm{peak}}$ at the transmitter.

### 3.2. Frequency Response of Transmitter and Receiver

Prior to measurement of higher harmonics, we investigate the frequency response of the LiN transmitter (5 MHz nominal center frequency and 0.5 inch dia.) in linear ultrasound range. The experiment was carried out using a broadband pulser and 1 cm thick Al block in the pulse-echo mode. A negative spike pulse was applied to the transmitting transducer to generate a pulsed ultrasound in the Al block. This input pulse was acquired from Panametrics 5052 pulser/receiver. Figure 2a shows the frequency spectrum of the pulse-echo test. The spectrum shows the first peak at 5 MHz which is the nominal center frequency of the transmit element or the fundamental resonance peak. The resonance peaks at about 15 MHz and 25 MHz correspond to the 3rd overtone and 5th overtone resonance frequencies, respectively. The resonance characteristic of the transmitter will be very effective for excitation of the fundamental wave at 5 MHz, and can also be used very effectively for selective and highly sensitive reception of third harmonic wave at approximately 15 MHz.

Next, we investigated the frequency response of the receive transducer (10 MHz nominal center frequency and 0.5 inch dia.) in linear ultrasound. The experiment was carried out in a similar manner to the transmit transducer. Figure 2b shows the frequency spectrum of the received signal. The spectrum shows a double-peak-frequency characteristic—a main lobe with the peak frequency at around 9 MHz and another minor lobe with the peak frequency at around 15 MHz. This type of transducer can be used to effectively transmit and receive 10 MHz signal, and can also be used as a receiver of about 15 MHz signal.

### 3.3. Received Output Signal and Magnitude Spectrum

The purpose of harmonics generation experiment was to obtain the frequency spectrum of the fundamental and harmonically generated waves, from which $\beta^{\prime }$ and $\gamma^{\prime }$ of each sample were obtained. Figure 3a,b show examples of the output signal and its Fourier spectrum of the 8 cm sample [24]. The second and third harmonic components are clearly seen at 2*f* = 10 MHz and 3*f* = 15 MHz, as well as the fundamental component at *f* = 5 MHz. The second harmonic is typically about 20 dB less than the magnitude of the fundamental while the third harmonic is typically about 20 dB less than the second harmonic. The amplitudes of the fundamental and harmonically generated waves are dependent on the input power level. 

### 3.4. Results of $\beta^{\prime }\;and\;\gamma^{\prime }$, and Combined Effect of Corrections

Figure 4a,b respectively show the results $\beta^{\prime }$ and $\gamma^{\prime }$ without and with the combined corrections for attenuation and diffraction. Here, $\beta^{\prime }$ of each sample was calculated from Equation (9) using the peak values of the magnitude spectrum $A_{1}$ and $A_{2}$. Similarly, $\gamma^{\prime }$ of each sample was calculated from Equation (10) using the peak values of the frequency spectrum $A_{1}$ and $A_{3}$. The influence of source nonlinearity correction will be considered in Section 3.6. The uncorrected $\beta^{\prime }$ shows a decreasing and then increasing behavior as the sample thickness increases. The combined effect of attenuation and diffraction corrections is found to shift the small and large values of $\beta^{\prime }$ closer to the average value. The overall behavior of $\beta^{\prime }$ after these corrections agrees well with the results of absolute β measurements [24]. 

The combined effect of attenuation and diffraction corrections on $\gamma^{\prime }$ is presented in Figure 4b. The uncorrected $\gamma^{\prime }$ values are rather uniform between 12 cm and 4 cm and increases suddenly from about 4 cm. The combined effect of attenuation and diffraction corrections is basically insignificant in the region of uniform $\gamma^{\prime }$, and the suddenly increasing behavior remains the same as the sample becomes shorter. 

Figure 4b shows both uncorrected and corrected $\gamma^{\prime }$ suddenly increase at 4 cm and shorter. This can be attributed to the large source nonlinearity compared to the small nonlinear third harmonic amplitude at short sample thicknesses. In contrast, Figure 4a shows the corrected $\beta^{\prime }$ does not exhibit such behavior and is very constant down to the shortest 2 cm sample, as there is no source nonlinearity included in the nonlinear second harmonic amplitude (see Figure 6b below). In the stable region of $\gamma^{\prime }$, the value of corrected $\gamma^{\prime }$ is much higer than the value of corrected $\beta^{\prime }$, which proves that the sensitivity of $\gamma^{\prime }$ is much better. Here, $\beta^{\prime }$ and $\gamma^{\prime }$ could be measured with the uncertainties of less than 5% and 10%, respectively. 

### 3.5. Comparison of $\gamma^{\prime }$ and ${\beta^{\prime }}^{2}$


The cubic nonlinear parameter ($\gamma^{\prime }=A_{3}/A_{1}^{3}$) is generally known to be more sensitive than the quadratic nonlinear parameter ($\beta^{\prime }=A_{2}/A_{1}^{2}$) for the same damage or defect. 

It is interesting to compare $\gamma^{\prime }$ with the square of $\beta^{\prime }$, ${\beta^{\prime }}^{2}$, based on the measured relative nonlinear parameters. Comparisons of uncorrected and corrected results of these parameters are presented in Figure 5a,b. Comparing $\gamma^{\prime }$ and ${\beta^{\prime }}^{2}$ of specimens with a thickness of 4 cm or longer, Figure 5a shows that the uncorrected $\gamma^{\prime }$ is smaller than the uncorrected ${\beta^{\prime }}^{2}$ except for the 4 cm specimen and the largest difference is about 50% at the 12 cm specimen. This difference becomes smaller as the sample thickness decreases. The corrected $\gamma^{\prime }$ and ${\beta^{\prime }}^{2}$ results are compared in Figure 5b. Since the combined effect of attenuation and diffraction corrections on $\gamma^{\prime }$ is very small, the overall trend between these two nonlinear parameters remains the same. The largest difference at the 12 cm specimen is almost the same. Based on these comparisons, it appears that the approximate relationship between the relative nonlinear parameters $\gamma^{\prime }$ and ${\beta^{\prime }}^{2}$ does not seem to hold. We will discuss this relationship further and draw a final conclusion since the effect of source nonlinearity correction was not taken into account here. 

### 3.6. Plot of $A_{1}^{3}$ vs. $A_{3}$ and Source Nonlinearity Check

According to Equation (10), $\gamma^{\prime }$ can be determined for each sample thickness *x* if the amplitudes of the fundamental and third harmonics are known. Figure 3b shows that the amplitudes of received higher harmonics are affected by the input power level. This also results in the dependence of $\gamma^{\prime }$ on the input power level. A reliable $\gamma^{\prime }$ for each sample can be determined by reducing the depence of $\gamma^{\prime }$ on the input power level. Another important thing is to examine and remove the presence of source nonlinearity in the received signal. In order to solve these problems together, Equation (10) is rearranged in a different form using the amplitude ratio $A_{3}\left(x\right)/A_{1}^{3}\left(x\right)$, which will depend on the input power level used in the experiment
(11)$${\left.\frac{A_{3}\left(x\right)}{A_{1}^{3}\left(x\right)}\right|}_{\begin{array}{c} input\\ power \end{array}}=\frac{\gamma^{\prime }k^{4}x^{2}}{32}\left[\frac{M_{3}\left(x\right)}{M_{1}^{3}\left(x\right)}\right]\left[\frac{D_{3}\left(x\right)}{D_{1}^{3}\left(x\right)}\right]$$

Equation (11) indicates that a plot of $A_{1}^{3}$ vs. $A_{3}$ obtained experimentally at different input power levels should follow a straight line with zero y-intercept. However, the actual plot may not be linear across all input power levels used. Furthermore, the y-intercept of the fitted line may not pass through the origin. The non-zero y-intercept indicates the amount of source nonlinearity included in the measurement system or the noise floor of the measurement system [25]. Therefore, the relationship plot of $A_{1}^{3}$ vs. $A_{3}$ obtained using the measurement data provides insight into the behavior of nonlinear solid samples and the stability of the measurement system against source nonlinearity. Similarly, a plot of $A_{1}^{2}$ vs. $A_{2}$ can be obtained for the second harmonic wave. 

The plot of $A_{1}^{3}$ vs. $A_{3}$ for the 8 cm sample is shown in Figure 6a. The open circles represent the measured data. The fitted straight line is shown together. The fitting was performed using the whole nine data points obtained at the nine input power levels used. There is a good linear relationship between these data. We can observe that $A_{3}$ contains a significant amount of source nonlinearity because the y-intercept is well above the origin. The reason for this was explained before. The source nonlinearity observed in the plot of $A_{1}^{3}$ vs. $A_{3}$ must be removed from $A_{3}$ for accurate determination of $\gamma^{\prime }$.

**Figure 6 materials-16-04739-f006:**
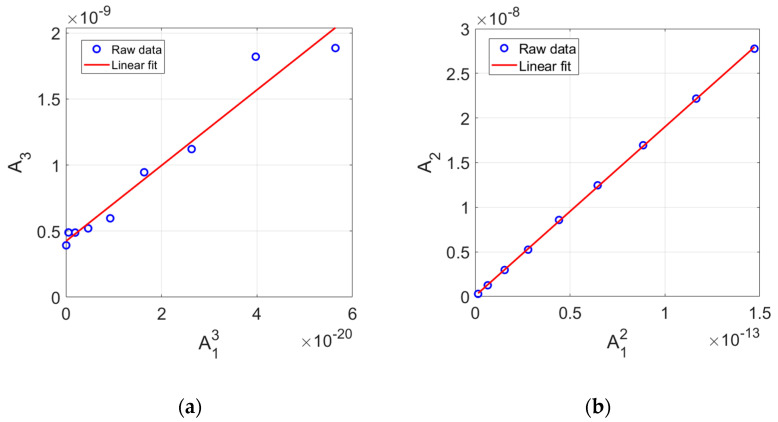
(**a**) Plot of $A_{1}^{3}$ vs. $A_{3}$, and (**b**) Plot of $A_{1}^{2}$ vs. $A_{2}$ at different input power levels used.

Similarly, the plot of $A_{1}^{2}$ vs. $A_{2}$ for the 8 cm sample is shown in Figure 6b. The same fitting scheme was used as before, and the best fit straight line is also shown. A very good linearity between these data is observed. The y-intercept almost goes through the origin, indicating that a relatively low noise floor and a very small amount of source nonlinearity was produced. Thus, the source nonlinearity which might be included in $A_{2}$ can be ignored. 

A plot of $A_{1}^{3}$ vs. $A_{3}$ was drawn for each sample, and the y-intercept was calculated by fitting all the measured data with a straight line. If source nonlinearity is found to exist, this source nonlinearity should be subtracted from the corresponding spectral peak value. Then, the combined effects of attenuation and diffraction were corrected according to Equation (10) to obtain all corrected $\gamma^{\prime }$. We focus here on the effect of source nonlinearity correction on $\gamma^{\prime }$. We also investigate the relationship between $\gamma^{\prime }$ and ${\beta^{\prime }}^{2}$ after source nonlinearity correction was mde for $A_{3}$. No source nonlinearity correction was mde for $A_{2}$. 

Figure 7a shows the comparison of $\gamma^{\prime }$ among three different corrections as a function of sample thickness—no correction, combined attenuation (M) and diffraction (D) correction, and total correction including the source nonlinearity correction and the combined M&D correction. The experimental data of each sample here were obtained at power level 30.

Combined M&D corrections shift the uncorrected $\gamma^{\prime }$ to a slightly higher or slightly lower value depending on the sample thickness, but the amount of corrections are very small especially in the 6 cm to 12 cm region. In contrast, additional source nonlinearity correction reduces the $\gamma^{\prime }$ value for all sample thicknesses, and the amount of reduction increases as the sample thickness becomes thinner. After the source nonlinearity correction, the stable measurement region of $\gamma^{\prime }$ now extends down to 4 cm, and the difference between $\gamma^{\prime }$ and ${\beta^{\prime }}^{2}$ in the stable 4 cm to 12 cm region becomes almost constant. However, the difference between these two parameters did not improve because the $\gamma^{\prime }$ value was further lowered after source nonlinearity correction. Comparing the average values of $\gamma^{\prime }$ and ${\beta^{\prime }}^{2}$ for the five samples in Figure 7a, the difference between these two parameters is about 56%. The maximum difference is about 63% for the 12 cm sample. 

Based on the investigation above, the difference between $\gamma^{\prime }$ and ${\beta^{\prime }}^{2}$ is found to be about 50% to 63% regardless of source nonlinearity correction. This difference is very large compared to the less than 10% difference between the absolute nonlinear parameters γ and $\beta^{2}$ [24]. Therefore, the approximate relationship between γ and $\beta^{2}$ cannot be confirmed through the measurement of relative nonlinear parameters, and the estimation of $\gamma^{\prime }$ through the measurement of $\beta^{\prime }$ or vice versa may not be accurate. Even after total correction was made, the $\gamma^{\prime }$ of the 2 cm sample is still very large. This could be attributed to the large source nonlinearity compared to the weak material-induced third harmonic generation. When the propagation distance or the sample thickness was insufficient, similar behavior was observed for the measured β of water and aluminum [25,26].

The behavior of $\gamma^{\prime }$ after total corrections is shown in Figure 7b as a function of input power level. Comparisons were made between $\gamma^{\prime }$ and ${\beta^{\prime }}^{2}$ of the 4 cm sample. Similar behaviors are expected for other samples. The values of $\gamma^{\prime }$ before total correction are extremely large because the combined effect of M&D correction is small. Unlike ${\beta^{\prime }}^{2}$, this trend of $\gamma^{\prime }$ gets worse at lower power levels. It can be seen that a relatively large amount of source nonlinearity is included in the amplitude of $A_{3}$ measured at lower input power levels. Therefore, source nonlinearity correction greatly reduces the values of uncorrected and M&D corrected $\gamma^{\prime }$. The effect of this correction becomes much more evident as the power level goes lower. The $\gamma^{\prime }$ after total correction has relatively uniform values down to a power level of about 10, and its differences from ${\beta^{\prime }}^{2}$ are almost constant.

Figure 7 clearly demonstrates the importance of confirming the presence of source nonlinearity in the received third harmonic amplitude and correcting it properly for accurate analysis of $\gamma^{\prime }$. 

## 4. Thickness Resonant Transducer for THG Measurement in Pulse-Echo Mode

Measurement techniques for $\beta^{\prime }$ and $\gamma^{\prime }$ of solids using longitudinal waves have been limited to through-transmission setups [13,14,15,16]. In terms of practical applications, the pulse-echo measurement technique is more desirable because of its many advantages. The traditional linear pulse-echo method, however, cannot be used for cumulative second harmonic generation and reliable measurements of $\beta^{\prime }$ for solid components [9,10,27,28]. Dual element transducers have been explored by the authors for the improvement of second harmonic generation in the pulse-echo NLUT of solids [15,20]. 

On the contrary, THG across the stress-free boundary of a solid is different from SHG. That is, the two third harmonic waves reflected from the stress-free boundary constructively interfere with each other, and the resulting third harmonic component accumulates as the propagation distance increases [11,12]. Therefore, it is possible to realize THG measurements in the pulse-echo mode. The amplitude of the third harmonic generated is equal to the amplitude generated by the through- transmission method whose propagation distance is twice the sample thickness.

To implement THG measurement in the pulse echo mode, a single crystal piezoelectric element with thickness resonance characteristics is preferred and applied in this section. Since the thickness resonant piezoelectric element has the frequency spectrum shown in Figure 2a, effective generation and highly sensitive reception of the third harmonic are possible. The measurement results of $\gamma^{\prime }$ are presented for a series of precipitation heat-treated samples with different aging times. We discuss the effects of source nonlinearity correction, variation of $\gamma^{\prime }$ as a function of aging time, and sensitivity comparison using the normalized $\gamma^{\prime }$. Our approach of using the thickness resonant transducer and source nonlinearity correction is considered to be one of the best solutions for pulse-echo third harmonic generation and measurement of the cubic nonlinear parameter. 

It is well known that the heat treatment of metal alloys causes a change in the microstructure of materials and thus a change in mechanical properties. One of the representative strengthening mechanisms is the precipitation hardening process through which extremely small, uniformly dispersed particles of a second phase form within the original phase matrix [29,30]. The fine particles of an impurity phase hinder the movement of dislocations and serve to harden the material. Since dislocations are the dominant mechanism of harmonic generation, the third harmonic will be generated differently according to microstructural changes with aging time [31,32]. Effects of precipitation on second harmonic generation in metals have been investigated experimentally [33,34] and theoretically [35,36,37].

### 4.1. Specimens and Experimental Setup

Aluminum 6061 alloy specimens with a thickness of 1 cm were prepared for heat treatment. The other dimensions of the specimens are 4 cm wide and 7 cm long. As shown in Figure 8, the specimens were subjected to a thermal cycle consisting of solution heat treatment and precipitation heat treatment [20]. First, all the specimens were solution heat treated at 540 °C for 4 h and then cooled in water for 2 h. After water cooling, artificial aging treatment was carried out for different times at a temperature of 220 °C. A total of 7 specimens were prepared: one right after water quenching (0 h) and 6 after artificial aging treatment with different times (1/3, 1/2, 1, 2, 48, 144 h).

Figure 9 shows the experimental setup for measurement of the harmonics generated by finite amplitude ultrasonic waves in pulse-echo mode. The transmit side is the same as the through-transmission method described in Section 3.1. One major difference is the use of a broadband diplexer (RDX-6, RITEC, Warwick, RI) to allow a single transducer to be used both for transmitting and receiving sound in pulse-echo testing. A single crystal lithium niobate (LiN) (5 MHz center frequency, 0.5 inch diameter) was used as a transmit/receive transducer (T/R), which was used as the transmitter in the through-transmission test of Section 3.1. The frequency response of this transducer is shown in Figure 2a. 

A high power toneburst of 5 MHz and 5 cycles was used as an input signal. For each sample, input powers of seven levels (from 0 to 60 power levels in 10 level steps) were applied from the high power pulser. These input voltages fall within the range of about 30–300 V_peak_ at the transmitter. 

### 4.2. Results of Uncorrected $\gamma^{\prime }$


Measurement of higher harmonics was carried out using the finite amplitude, pulse-echo method. The purpose of the experiment was to acquire $A_{1}$ and $A_{3}$, the peak values of frequency spectrum at the fundamental and third harmonic frequencies, from which $\gamma^{\prime }$ of each sample was determined. Figure 10a,b show the electrical output signal and its Fourier spectrum measured on the #1 sample at the input power level 40 when 5 cycles of sine wave toneburst was applied. As shown in Figure 10b, in addition to the spectral peak at the fundamental frequency of *f* = 5 MHz, the spectral peaks near the 3*f* and 5*f* frequencies are clearly visible. The generation of higher harmonic peaks here are due to overtone thickness resonance of the LiN piezoelectric element. Because the LiN element was not tuned and used as fabricated, the higher harmonic peaks do not occur precisely at 15 MHz and 25 MHz. It is also noted that the generated third harmonic peak is about 40 dB lower than that of the fundamental wave. Because of the destructive interference of the second harmonics after being reflected from the stress-free surface, the received second harmonic is negligibly small, and the pulse-echo technique cannot be used for a reliable measurement of $\beta^{\prime }$. The magnitude of the received signal in the time and frequency domains is directly affected by the applied input power level.

The relative cubic nonlinear parameter of each sample at a specific input power level was calculated using $\gamma^{\prime }=A_{3}/A_{1}^{3}$, where $A_{1}$ and $A_{3}$ are the spectral peak values at the fundamental and third harmonic frequencies, respectively. Figure 11 shows $\gamma^{\prime }$ of seven samples measured at seven different input power levels. The calculation of these parameters is based on the relationship between $A_{1}^{3}$ and $A_{3}$ measured at seven different input power levels shown in Figure 11b. The results of $\gamma^{\prime }$ in Figure 11a are before the source nonlinearity correction was made. The detailed procedure of source nonlinearity correction is described in the next section, and the correlation between the source nonlinearity-corrected $\gamma^{\prime }$ and the aging time will be discussed.

### 4.3. Source Nonlinearity Correction

A plot of $A_{1}^{3}$ vs. $A_{3}$ can be drawn for different input power levels to check the noise floor of measurement system and/or the amount of source nonlinearity included in $A_{3}$. Figure 11b shows plots of $A_{1}^{3}$ vs. $A_{3}$ for all seven samples at seven input power levels used. 

Looking at Figure 11b, it can be seen that the graph of $A_{1}^{3}$ vs. $A_{3}$ consists of approximately three linear regions with different slopes as the input voltage increases: a steep slope at very low input voltages below PL10, a moderate slope at intermediate input voltages from PL10 to PL50, and a relatively steep slope again at high input voltages above PL50. The $A_{1}^{3}$ vs. $A_{3}$ relation curve seems to depend on the type of transducers being used, as will be further discussed later. There is no strict criterion for linearization process, but in this study the data of the second region was used for linear curve fitting to obtain the slope and y-intercept. Since the samples are not thick and the combined effect of diffraction and attenuation corrections are insignificant as shown in Section 3.4, only source nonlinearity corrections were taken into account here.

Figure 12a shows the raw data used in the fitting and the results of the best fit straight line. A good linearity is observed between these data in each sample. It is clearly seen that the y-intercept is well above the origin, indicating that a significant amount of source nonlinearity is included in the measured $A_{3}$. The y-intercept of each sample was subtracted from $A_{3}$ of that sample, and the $\gamma^{\prime }$ after source nonlinearity correction was calculated. The results are shown in Figure 12b.

Comparing the results of $\gamma^{\prime }$ before and after source nonlinearity correction (Figure 11a and Figure 12b), $\gamma^{\prime }$ before correction is widely scattered over the five input power levels used for linear curve fitting, whereas $\gamma^{\prime }$ after correction is very narrowly clustered. The dependence of the corrected $\gamma^{\prime }$ of each sample on the input power is now greatly reduced, and this decrease in input voltage dependence is more evident at low input voltages. In addition, after correction for source nonlinearity, the value of $\gamma^{\prime }$ was decreased in all samples and the characteristic behavior as a function of the aging time was maintained, so the correction appears to have been made adequately.

Figure 12b shows the variation of corrected $\gamma^{\prime }$ as a function of aging time. Right after solution heat treatment and water quenching at 220 °C, $\gamma^{\prime }$ shows a continuous decrease until 20 min. of aging time and then a slight increase at 40 min. and a decrease again to reach a minimum at about one hour. After reaching the minimum value, it increases rapidly at about 2 h, then rapidly decreases at 48 h, and gradually increases at 144 h. This behavior of $\gamma^{\prime }$ agrees very well with that of the absolute β measurement results [20], but their sensitivity is different as compared in the next section. The variation of $\gamma^{\prime }$ as a function of the aging time can be explained by the microstructural change of the material due to the generation, evolution, and extinction of precipitates inside the specimen caused by the precipitation heat treatment at different aging times [38,39,40].

### 4.4. Comparison of Sensitivity and Overall Discussion

The absolute quadratic nonlinear parameter β, is measured in the previous study for the same heat-treated samples using a dual element transducer and the pulse-echo method [20]. In this section, the normalized $\gamma^{\prime }$ after source nonlinearity correction is compared with the normalized β. In addition, the normalized $\gamma^{\prime }$ is compared with the normalized $\beta^{2}$. The average value of $\gamma^{\prime }$ obtained from the five input power levels shown in Figure 12b is used here as the $\gamma^{\prime }$ value of each sample. The values of #1 sample are used for normalization in all three comparisons. The comparison results are shown in Figure 13. The normalized $\gamma^{\prime }$ and β shows a very similar behavior to each other in that the minimum occurs after one hour and the maximum occurs after two hours of aging time. The normalized $\beta^{2}$ also shows a similar behavior. 

If we compare the sensitivity, the normalized $\gamma^{\prime }$ shows better sensitivity than the normalized β and $\beta^{2}$ over the entire aging times. In particular, the normalized $\gamma^{\prime }$ at the aging times of one hour and two hours, where all three normalized parameters have minimum and maximum values, shows much better sensitivity than the normalized β and $\beta^{2}$. The excellent sensitivity of $\gamma^{\prime }$ shown here is attributed to the thickness resonant transducer that can effectively generate and sensitively detect the fundamental and third harmonics.

In NLUT, it is necessary to check for the presence of source nonlinearity in the measured higher harmonic amplitudes. In particular, when a piezoelectric transmit transducer is used to generate odd harmonics of the fundamental, such as the third or fifth harmonics, it is essential to check and eliminate the source nonlinearity since some degree of source nonlinearity is always accompanied [24]. The source nonlinearity correction technique proposed in this study is easy to apply and has practical advantages because it uses the linearity of experimental data that is usually checked during harmonics generation measurement. 

The behavior of $A_{1}^{3}$ vs. $A_{3}$ seems to depend on the type of transducers and input voltage levels used in the harmonic generation measurement. In this study, two types of transmit/receive transducers were used and their $A_{1}^{3}$ vs. $A_{3}$ behavior showed a slight difference. They are (i) the bare crystal transmitter and the double-peak-frequency type receiver for the through-transmission testing, and (ii) the bare crystal transmitter/receiver for the pulse-echo testing. The same bare crystal was used as a transmitter in these tests. In the first case, as shown in Figure 6b, all data in the entire pressure range were used for linear curve fitting. In the second case, as shown in Figure 11b, the data in the middle pressure range except for very low and very high pressures were used for linear curve fitting. The selection of an appropriate pressure range for linear curve fitting is very important as it directly affects the resulting value of $\gamma^{\prime }$. It seems necessary to develop a more systematic method for this. 

Piezoelectric elements or transducers showing a good characteristic of thickness resonances at odd-numbered harmonic frequencies can be very effective in actuating and sensing the fundamental and third harmonic waves in THG measurement. Without using backing materials or with using minimal backing materials, such thickness resonant transducers or elements can have high sensitivity transmit/receive potential in the pulse-echo mode. Their narrow bandwidths at the fundamental and third harmonic frequencies are very effective in generating these frequency waves under finite amplitude toneburst excitation, and in receiving them with good selectivity and high sensitivity. The excitation and reception sensitivity at specific resonance frequencies can be improved through overtone polishing or electrical tuning [21]. Thickness resonant transducers that are sensitive to the fifth harmonic frequency can be used for the pulse-echo fifth harmonic generation and the measurement of quintic nonlinear parameters.

## 5. Conclusions

In this paper, the measurement of relative cubic nonlinear parameters through the generation of third harmonics in the pulse-echo mode was addressed. To this end, the importance of a single crystal thickness resonant transducer that satisfies the requirements of transmitter and receiver in the pulse-echo NLUT was emphasized. In addition, the method of checking and correcting the source nonlinearity that unavoidably appears during the measurement of third harmonic amplitude was dealt with great importance. In through-transmission measurements of $\gamma^{\prime }$ on a series of aluminum samples of different thicknesses, the combined effect of attenuation and diffraction correction on the measured $\gamma^{\prime }$ was negligibly small, but source nonlinearity correction has a more pronounced effect. Due to source nonlinearity correction, $\gamma^{\prime }$ could be accurately determined at thinner samples and lower input voltages. The approximate relationship of $\gamma \approx \beta^{2}$ between γ and β squared in the measurement of absolute nonlinear parameters has not been identified in the measurement of relative nonlinear parameters, thus the estimation of $\beta^{\prime }$ from the measured $\gamma^{\prime }$ cannot be made accurately.

For a series of precipitation heat-treated samples, the pulse-echo method using a thickness resonant transducer sensitively detected $\gamma^{\prime }$ representing changes in microstructure due to different aging times. Checking and correcting the source nonlinearity greatly reduced the dependence of $\gamma^{\prime }$ on the input voltage, and the effect was noticeable at low input voltages. As a result, the final $\gamma^{\prime }$ could be obtained with an error of less than 10% in all heat-treated samples. The sensitivity of the three nonlinear parameters—$\gamma^{\prime }$, β and $\beta^{2}$—was compared with each other using normalized nonlinear parameters. As expected, $\gamma^{\prime }$ showed the best sensitivity.

The sensitivity of a transmit/receive transducer that generates and receives both fundamental and third harmonics depends on the center frequency of the transducer’s bandwidth and its strength. Since thickness resonant transducers operate very sensitively around these frequencies as fabricated, they can easily meet the frequency bandwidth requirements as sensitive transmitters and receivers. Improving the transducer’s sensitivity at higher harmonics will make the damage detection more sensitive in the pulse-echo mode through the generation and reception of superharmonics such as the fifth harmonic. The proposed source nonlinearity correction technique is expected to be more frequently used in the future because it has many practical advantages that can accurately measure $\gamma^{\prime }$ without adding hardware to the measurement system.

## Figures and Tables

**Figure 1 materials-16-04739-f001:**
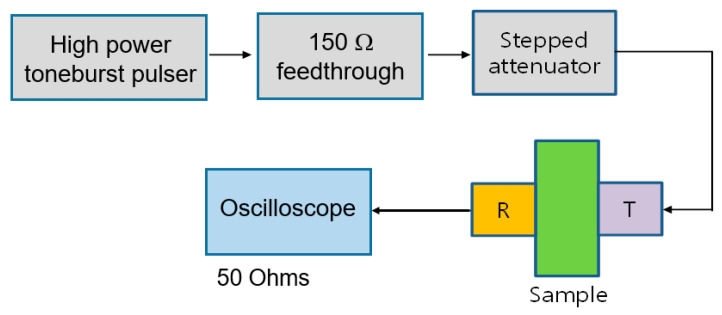
Schematic of the experimental setup for harmonics generation and reception in through-transmission mode.

**Figure 2 materials-16-04739-f002:**
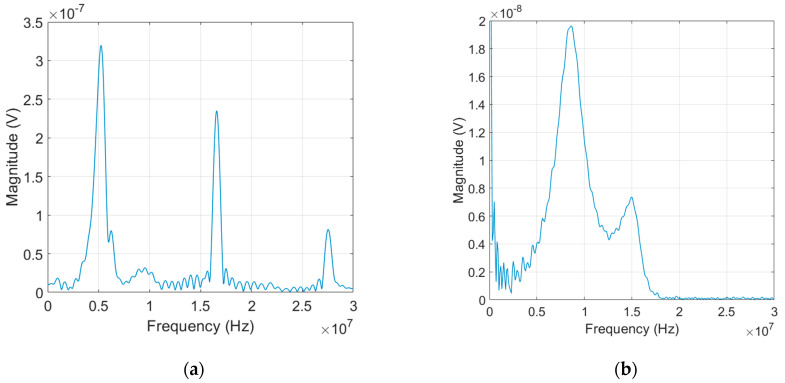
Frequency response of (**a**) LiN transmit element with a nominal center frequency of 5 MHz, and (**b**) receive transducer with a nominal center frequency of 10 MHz.

**Figure 3 materials-16-04739-f003:**
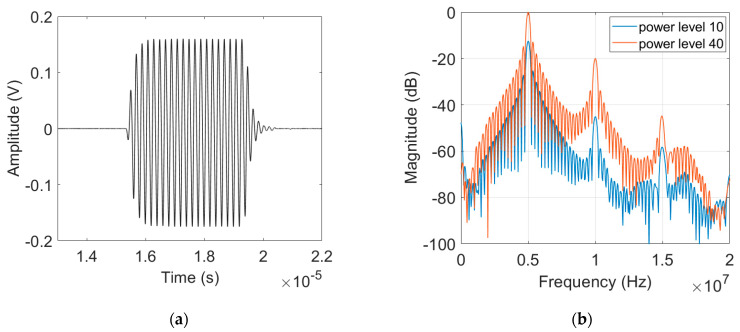
Received output signal from the 8 cm sample: (**a**) time domain waveform, and (**b**) magnitude spectrum.

**Figure 4 materials-16-04739-f004:**
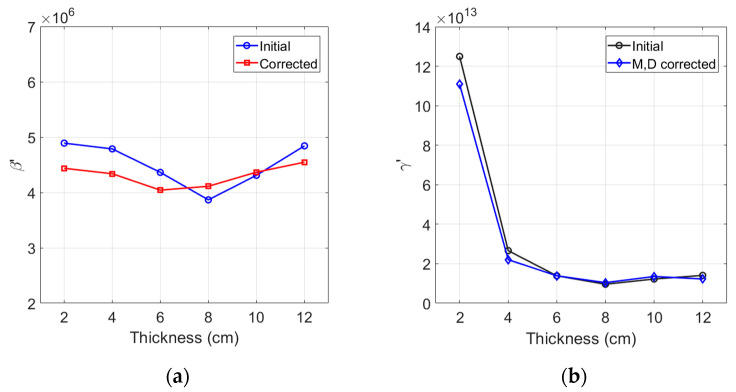
Combined effect of attenuation and diffraction corrections on (**a**) $\beta^{\prime }$, and (**b**) $\gamma^{\prime }$.

**Figure 5 materials-16-04739-f005:**
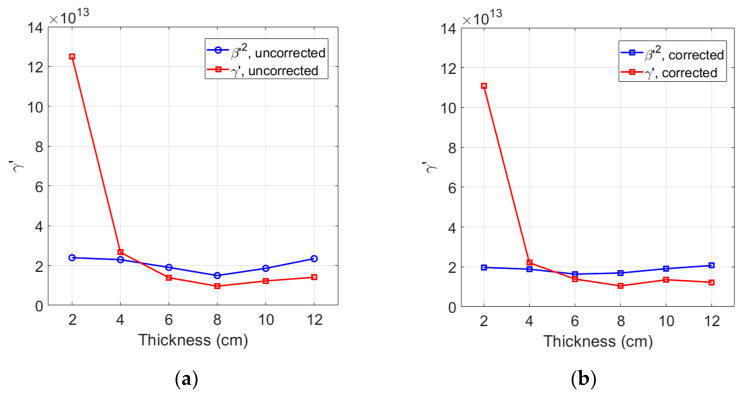
Comparison of $\gamma^{\prime }$ with ${\beta^{\prime }}^{2}$: (**a**) without, and (**b**) with combined corrections for attenuation and diffraction.

**Figure 7 materials-16-04739-f007:**
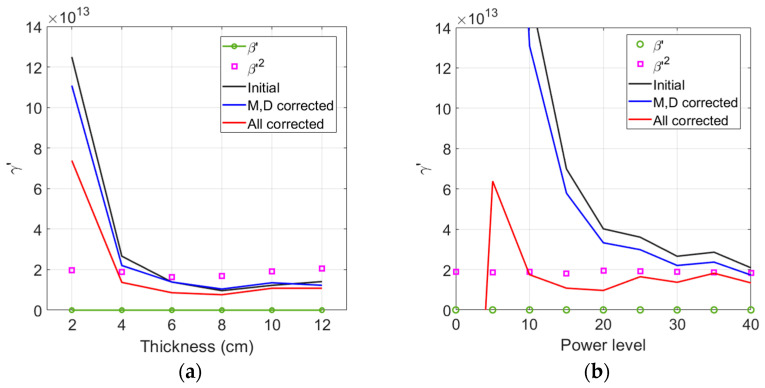
Effect of total correction on $\gamma^{\prime }$ and comparison with ${\beta^{\prime }}^{2}$ as a function of sample thickness (**a**), and input power level (**b**).

**Figure 8 materials-16-04739-f008:**
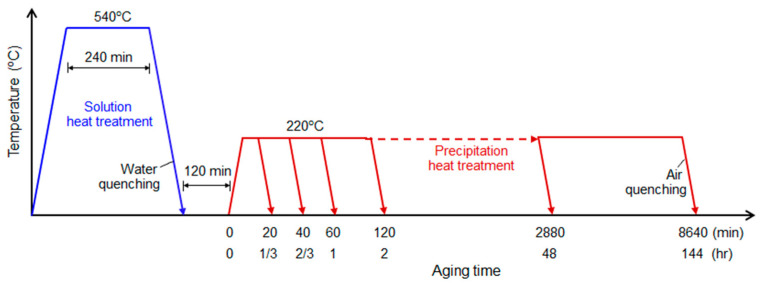
Schematic diagram for precipitation heat treatment of aluminum 6061 alloy.

**Figure 9 materials-16-04739-f009:**
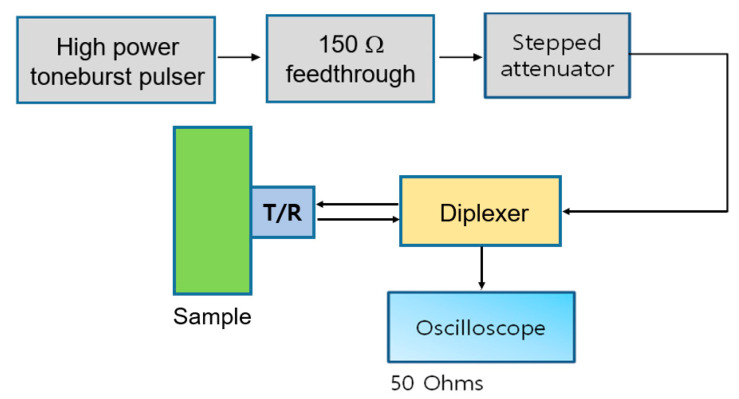
Schematic of the experimental setup for measurement of the harmonics generated by finite amplitude ultrasonic waves in pulse-echo mode.

**Figure 10 materials-16-04739-f010:**
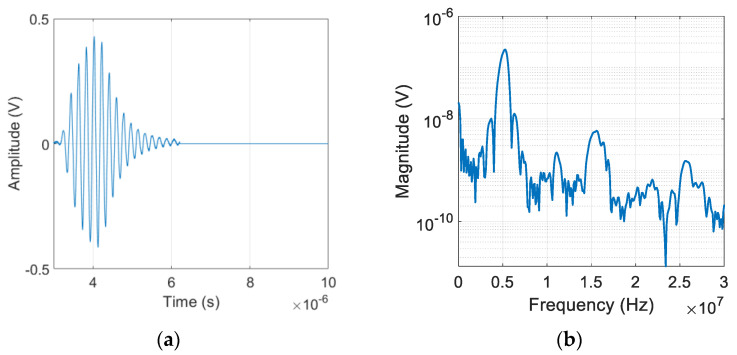
Received output signal from #1 sample: (**a**) typical time domain waveform, and (**b**) magnitude spectrum.

**Figure 11 materials-16-04739-f011:**
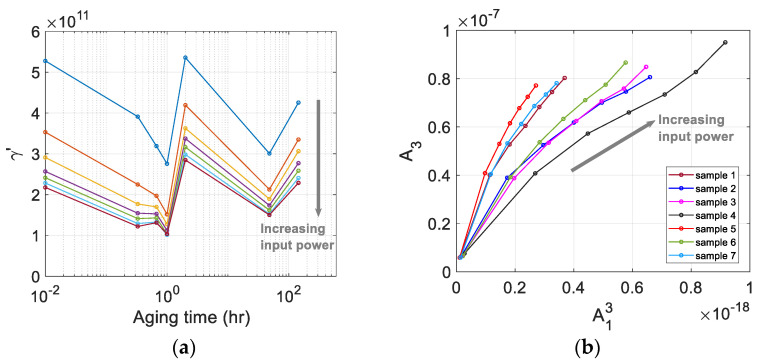
(**a**) The uncorrected cubic nonlinear parameter $\gamma^{\prime }$, and (**b**) the plot of $A_{1}^{3}$ vs. $A_{3}$ of seven samples measured at seven different input power levels.

**Figure 12 materials-16-04739-f012:**
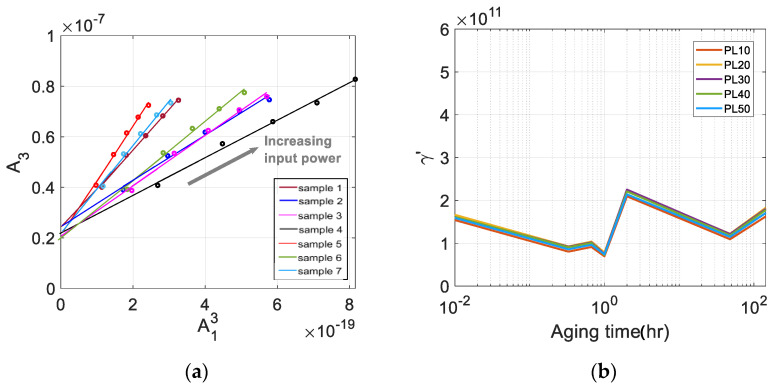
(**a**) The best fit straight line for the plot of $A_{1}^{3}$ vs. $A_{3}$ measured at five input power levels, and (**b**) the calculated $\gamma^{\prime }$ of seven samples after source nonlinearity correction.

**Figure 13 materials-16-04739-f013:**
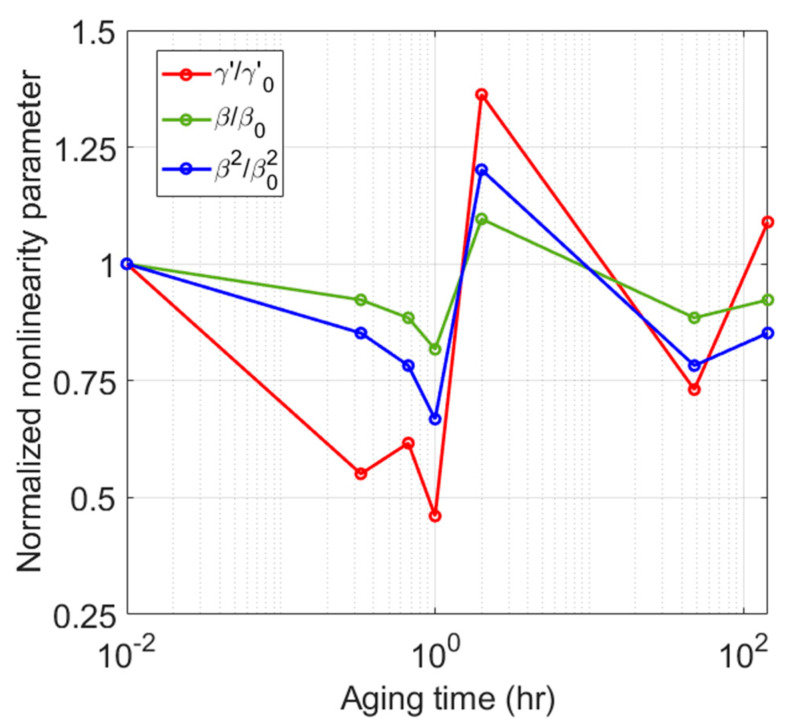
Comparison of sensitivity using normalized nonlinear parameters.

## Data Availability

The data presented in this study are available on request from the corresponding author.

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
