# Peer review of "Highly Sensitive Detection of Microstructure Variation Using a Thickness Resonant Transducer and Pulse-Echo Third Harmonic Generation"

_materials, 2023, doi:10.3390/ma16134739_

Round 1

Reviewer 1 Report

The author present a pulse-echo method to measure the cubic nonlinearity. The authors show the effectiveness of PE measured gamma for material characterization. While the manuscript is of interest to the community, there are several fundamental questions that remain unanswered. 

1. Line 97: The phase reversal is caused in the fundamental harmonic, which causes harmonic generation to be out of phase between the forward and backward paths. The authors claim the third harmonic does not reverse in phase, which is not true. Any incident wave (of any frequency) will reverse in phase due to the stress-free boundary, purely due to the reflection coefficient. This raises question on monotonic accumulation of 3rd harmonic. Unless the authors can first prove this, rest of the manuscript is not relevant. I invite the authors to provide evidence of this phenomenon or provide relevant citations. 

2. Fig. 1: is it a 150 ohm feedthrough or 50 ohm feedthrough?

3. Fig. 11: I recommend using thinner line or different types of lines to show the trend. The figure looks very messy. 

4. The authors present the importance of diffraction of attenuation corrections, but dont present it for aged samples. The effect of attenuation is actually apparent in Fig. 12(a) where the A1 values are different for each sample. Without attenuation correction, it is impossible to determine the true effect. 

Language can be improved in several places. There are several missing words, grammatical mistakes and sentence formation issues. I recommend a close edit to remove these mistakes. 

Author Response

Authors’ Reply to Reviewer 1

The authors are very grateful for the valuable comments from the reviewer. We checked the review results very carefully and used our best knowledge to answer all the comments and questions. The paper has been revised as much as possible to reflect the reviewer's opinion.

The revised part of the paper is marked with blue letters in the manuscript.

The author present a pulse-echo method to measure the cubic nonlinearity. The authors show the effectiveness of PE measured gamma for material characterization. While the manuscript is of interest to the community, there are several fundamental questions that remain unanswered.

  1. Line 97: The phase reversal is caused in the fundamental harmonic, which causes harmonic generation to be out of phase between the forward and backward paths. The authors claim the third harmonic does not reverse in phase, which is not true. Any incident wave (of any frequency) will reverse in phase due to the stress-free boundary, purely due to the reflection coefficient. This raises question on monotonic accumulation of 3rd harmonic. Unless the authors can first prove this, rest of the manuscript is not relevant. I invite the authors to provide evidence of this phenomenon or provide relevant citations.

The second harmonic generation process in the pulse-echo mode was added based on our extensive previous studies. The third harmonic generation process in the pulse-echo mode was described qualitatively. The latest paper on third harmonic generation in the pulse-echo mode was added in Ref. [11]:

The second harmonic behavior at the stress-free reflecting boundary can be described as follows. According to the quasilinear theory of nonlinear wave equations [9,10], the solution for the second harmonic in the reflected beam consists of two separate contributions. One is the reflected second harmonic wave which is generated in the forward propagation, while the other one is the newly generated by the reflected fundamental wave in the backward propagation. Because of the π phase difference between these two components, the second harmonic completely cancels out after reflection from the stress-free boundary. For this reason, most second harmonic generation measurements have been limited to through-transmission methods. 

Unlike the behavior of second harmonics [9,10], the behavior of the third harmonic at the stress-free boundary can be treated like a continuous transmission wave propagating twice the sample thickness because there is no phase difference between the two third harmonic components after reflection [11,12]. The resulting third harmonic component accumulates in proportion to the propagation distance. Therefore, the pulse-echo THG measurement can be realized.

  1. Fig. 1: is it a 150 ohm feedthrough or 50 ohm feedthrough?

150 ohm feedthrough is correct. I have checked with RITEC Inc. The main reason is to be used with RPR4000 high power amplifier which has 8 kW nominal output power.

50 ohm feedthrough can be used for GA2500A, for instance, which has lower nominal output power of 5 kW.

  1. Fig. 11: I recommend using thinner line or different types of lines to show the trend. The figure looks very messy.

Figure 11 was changed as recommended.

  1. The authors present the importance of diffraction of attenuation corrections, but dont present it for aged samples. The effect of attenuation is actually apparent in Fig. 12(a) where the A1 values are different for each sample. Without attenuation correction, it is impossible to determine the true effect.

The purpose of the paper is (i) to develop a third harmonic measurement technique in the pulse-echo mode using a thickness resonant transducer, and (2) to develop a source nonlinearity correction method. Therefore, the importance of the thickness resonant transducer was emphasized. In addition, the procedure of checking and correcting source nonlinearity was emphasized throughout the paper.

The combined effect of attenuation and diffraction correction was considered in the first part of experiment for AL samples. However, this effect was not taken into account in the second part of experiment for precipitation heat-treated samples. The reason for this is as follows:

(1) The combined effect of attenuation and diffraction correction on was very small even for very long samples, as shown in Fig. 4(b).

(2) In Fig. 12(a), all 7 samples have the same thickness, 1 cm, but have different aging times.

Therefore, even the combined attenuation and diffraction correction will have almost the same effect on these samples, and will not change Fig. 12(b), the variation of  as a function of time.

  1. Language can be improved in several places. There are several missing words, grammatical mistakes and sentence formation issues. I recommend a close edit to remove these mistakes.

English has been checked thoroughly, and corrected/modified as best as possible.

Reviewer 2 Report

Comments:

1- In abstract the author must provide the highlight of this study

2. What are the advantages of employed technique?

3. What software or coding program was used to obtain the measurement results ?

4- The authors should try to give advantageous of using of their method compared to others.

5 The authors need to explain that their method used in the research is one of the appropriate solutions in the context of the research problem. What are the achievements of previous studies based on their current method? Also, describe what has not been achieved?

6 There are still many papers in the allied fields that are not cited in introduction section. It is understandable to colleagues from a broad range of scientific disciplines. For that purpose, I would like request author to add the following recent papers, which can help to enhance the introduction section:

https://doi.org/10.1109/JSEN.2020.3006376

https://doi.org/10.3390/polym14142883

https://doi.org/10.3390/fractalfract7030214

7. There are many research papers study the same problem which investigated in the present paper. What is exactly the new point of this work?

8. The conclusions should be clear and re-written; it must be enriched about discussion on solving problem

Author Response

Authors’ Reply to Reviewer 2

The authors of this paper are very grateful for the valuable comments from the reviewer. We took the review results very carefully and used our best knowledge to answer all the comments and questions. The paper has been revised as much as possible to reflect the reviewer's opinion.

The revised part of the paper is marked with blue letters in the manuscript.

Comments:

  1. In abstract the author must provide the highlight of this study

The purpose of this paper is twofold: (i) to develop a third harmonic measurement technique in the pulse-echo mode using a thickness resonant transducer, and (2) to develop a source nonlinearity correction method. These purposes are well highlighted in the abstract:

Abstract: In nonlinear ultrasound testing, the relative nonlinearity parameter is conveniently measured as a sensitive means of detecting and imaging overall variation of microstructures and damages. Compared to the quadratic nonlinearity parameter (β'), the cubic nonlinearity parameter (γ'), calculated as the third harmonic amplitude divided by the cube of the fundamental amplitude, has generally a higher value, providing better sensitivity in nonlinearity parameter mapping. Since the third harmonic amplitude is about two orders of magnitude lower than the fundamental amplitude, efficient excitation and highly sensitive reception of third harmonic is very important. In this paper, we explore an odd harmonic thickness resonant transducer that meets the requirements for pulse-echo third harmonic generation (THG) measurements. We also address the problem of source nonlinearity that may be present in the measured third harmonic amplitude and propose a method to properly correct it. First, we measure γ' for a series of aluminum specimens using the through-transmission method to observe the behavior of γ' as a function of specimen thickness and input voltage, and examine the effects of various corrections such as attenuation, diffraction and source nonlinearity. Next, we apply the odd harmonic resonant transducer to pulse-echo THG measurements of precipitation heat-treated specimens. It is shown that such transducer is very effective in generation and detection of fundamental and third harmonics under finite amplitude toneburst excitation. The highly sensitive detection results of γ' are presented as a function of aging time, and the sensitivity of γ' is compared with that of β' and .

  1. What are the advantages of employed technique?

The advantages of employed technique are (i) use of a thickness resonant transducer that is effective in exciting and receiving the fundamental and third harmonic waves, (ii) pulse-echo measurement that has great advantages in field applications, (iii) cubic nonlinearity measurement that provides high sensitivity in damage detection, (iv) source nonlinearity correction method that is easy to apply for accurate measurement of cubic nonlinearity parameter. These advantages are well described in the introduction, conclusions, and appropriate places of the paper.

  1. What software or coding program was used to obtain the measurement results ?

Matlab program was used to perform signal processing and to obtain the measurement results.

  1. The authors should try to give advantageous of using of their method compared to others.

Please refer to answers in No. 2 above.

  1. The authors need to explain that their method used in the research is one of the appropriate solutions in the context of the research problem. What are the achievements of previous studies based on their current method? Also, describe what has not been achieved?

(i) Our method or approach used in this work is one of the appropriate solutions: use of thickness resonance transducer and source nonlinearity correction for pulse-echo third harmonic generation and measurement of the cubic nonlinearity parameter.

(ii) The current method is only proposed in this paper, and has never been used/applied before. However, regarding the pulse-echo nonlinear ultrasound method for second harmonic generation and measurement of the absolute second-order (quadratic) nonlinearity parameter b, we have developed a dual element transducer method with diffraction and attenuation corrections. Although the dual element transducer method can be developed further to be used for the pulse-echo third harmonic generation and measurement of the absolute or relative cubic nonlinearity parameter g or g’. However, the thickness resonant transducer is better than the dual element transducer because the former is a single transducer, easy to fabricate and handle, and more sensitive. Therefore, we decided to use the thickness resonant transducer.  

(iii) The objectives and contents of this paper are what have not been achieved, therefore we would like to achieve.

The introductory part of Section 4 provides authors’ explanation to Q5:

Measurement techniques for β' and γ' of solids using longitudinal waves have been limited to through-transmission setups [13–16]. For practical applications, pulse-echo measurement techniques that enable single-sided access to test components are more desirable. However, when using the second harmonic wave reflected from the stress-free boundary, it is difficult to obtain a reliable β' because such boundary destructively changes the second harmonic generation process [9,10,27]. A dual element transducer has been proposed by the authors to improve second harmonic generation in the pulse-echo NLUT of solids [15,20].

In contrast, third harmonic generation at the stress-free boundary is different from second harmonic generation. That is, the two third harmonic waves reflected from the stress-free boundary constructively interfere with each other, and the resulting third harmonic component accumulates as the propagation distance increases. Therefore, it is possible to realize THG measurements in the pulse-echo mode. The amplitude of the third harmonic generated is equal to the amplitude generated by the through- transmission method whose propagation distance is twice the sample thickness.

To implement THG measurement in the pulse echo mode, a single crystal piezoelectric element with thickness resonance characteristics is preferred and applied in this section. Since the thickness resonant piezoelectric element has the frequency spectrum shown in Fig. 2(a), effective generation and highly sensitive reception of the third harmonic are possible. The measurement results of γ' are presented for a series of precipitation heat-treated samples with different aging times. We discuss the effects of source nonlinearity correction, variation of γ' as a function of aging time, and sensitivity comparison using the normalized γ'. Our approach of using the thickness resonant transducer and source nonlinearity correction is considered to be one of the best solutions for pulse-echo third harmonic generation and measurement of the cubic nonlinearity parameter.

  1. There are still many papers in the allied fields that are not cited in introduction section. It is understandable to colleagues from a broad range of scientific disciplines. For that purpose, I would like request author to add the following recent papers, which can help to enhance the introduction section:

https://doi.org/10.1109/JSEN.2020.3006376

Optimization and Validation of Dual Element Ultrasound Transducers for Improved Pulse-Echo Measurements of Material Nonlinearity

=> This reference is directly related to the current work, so included as Ref. [20].

https://doi.org/10.3390/polym14142883

Three-Dimensional Boundary Element Strategy for Stress Sensitivity of Fractional-Order Thermo-Elastoplastic Ultrasonic Wave Propagation Problems of Anisotropic Fiber-Reinforced Polymer Composite Material

=> This reference is not related to the current work, so not included.

https://doi.org/10.3390/fractalfract7030214

A Nonlinear Fractional BEM Model for Magneto-Thermo-Visco-Elastic Ultrasound Waves in Temperature-Dependent FGA Rotating Granular Plates

=> This reference is not related to the current work, so not included.

  1. There are many research papers study the same problem which investigated in the present paper. What is exactly the new point of this work?

No. That is not true. No papers in the literature investigate the same problem as in the present paper.

We could not find any paper that uses thickness resonant transducers to measure cubic nonlinearity parameters in the pulse-echo mode with source nonlinearity correction.

Research papers on third harmonic generation measurement due to microdamage and precipitation were added in the text:

Measurements of γ' for fatigue cracks [1,2], plastic deformation [3-5] and microstructures [6-8] have been reported in the literature.

Effects of precipitation on second harmonic generation in metals have been investigated experimentally [32,33] and theoretically [34-36].

  1. The conclusions should be clear and re-written; it must be enriched about discussion on solving problem.

The conclusions were completely rewritten:

In this paper, the measurement of relative cubic nonlinearity parameters through the generation of third harmonics in the pulse-echo mode was addressed. To this end, the importance of a single crystal thickness resonant transducer that satisfies the requirements of transmitter and receiver in the pulse-echo NLUT was emphasized. In addition, the method of checking and correcting the source nonlinearity that unavoidably appears during the measurement of third harmonic amplitude was dealt with great importance. In through-transmission measurements of γ' on a series of aluminum samples of different thicknesses, the combined effect of attenuation and diffraction correction on the measured γ' was negligibly small, but source nonlinearity correction has a more pronounced effect. Due to source nonlinearity correction, γ' could be accurately determined at thinner samples and lower input voltages. The approximate relationship of γ≈β^2 between γ and β squared in the measurement of absolute nonlinearity parameters has not been identified in the measurement of relative nonlinearity parameters, thus the estimation of β' from the measured γ' cannot be made accurately.

For a series of precipitation heat-treated samples, the pulse-echo method using a thickness resonant transducer sensitively detected γ' representing changes in microstructure due to different aging times. Checking and correcting the source nonlinearity greatly reduced the dependence of γ' on the input voltage, and the effect was noticeable at low input voltages. As a result, the final γ' could be obtained with an error of less than 10% in all heat-treated samples. The sensitivity of the three nonlinearity parameters – γ', β and β^2 – was compared with each other using normalized nonlinearity parameters. As expected, γ' showed the best sensitivity.

The sensitivity of a transmit/receive transducer that generates and receives both fundamental and third harmonics depends on the center frequency of the transducer's bandwidth and its strength. Since thickness resonant transducers operate very sensitively around these frequencies as fabricated, they can easily meet the frequency bandwidth requirements as sensitive transmitters and receivers. Improving the transducer’s sensitivity at higher harmonics will make the damage detection more sensitive in the pulse-echo mode through the generation and reception of superharmonics such as the fifth harmonic. The proposed source nonlinearity correction technique is expected to be more frequently used in the future because it has many practical advantages that can accurately measure γ' without adding hardware to the measurement system.
